# Lentiviral Vectors as Tools for the Study and Treatment of Glioblastoma

**DOI:** 10.3390/cancers11030417

**Published:** 2019-03-24

**Authors:** Claudia Del Vecchio, Arianna Calistri, Cristina Parolin, Carla Mucignat-Caretta

**Affiliations:** Department of Molecular Medicine, University of Padova, 35121 Padova, Italy; claudia.delvecchio@unipd.it (C.D.V.); arianna.calistri@unipd.it (A.C.); cristina.parolin@unipd.it (C.P.)

**Keywords:** glioblastoma, lentiviral vectors, gene therapy, animal models

## Abstract

Glioblastoma (GBM) has the worst prognosis among brain tumors, hence basic biology, preclinical, and clinical studies are necessary to design effective strategies to defeat this disease. Gene transfer vectors derived from the most-studied lentivirus—the Human Immunodeficiency Virus type 1—have wide application in dissecting GBM specific features to identify potential therapeutic targets. Last-generation lentiviruses (LV), highly improved in safety profile and gene transfer capacity, are also largely employed as delivery systems of therapeutic molecules to be employed in gene therapy (GT) approaches. LV were initially used in GT protocols aimed at the expression of suicide factors to induce GBM cell death. Subsequently, LV were adopted to either express small noncoding RNAs to affect different aspects of GBM biology or to overcome the resistance to both chemo- and radiotherapy that easily develop in this tumor after initial therapy. Newer frontiers include adoption of LV for engineering T cells to express chimeric antigen receptors recognizing specific GBM antigens, or for transducing specific cell types that, due to their biological properties, can function as carriers of therapeutic molecules to the cancer mass. Finally, LV allow the setting up of improved animal models crucial for the validation of GBM specific therapies.

## 1. Introduction

Glioblastoma (GBM) is the most aggressive brain tumor, with poor prognosis and scarce progression-free survival time, usually leading to death within 20 months after initial diagnosis [1]. Currently, GBM classification by the World Health Organization couples clinical features to molecular phenotype, distinguishing isocitrate dehydrogenase (IDH) wildtype GBMs, mostly primary developed tumors accounting for nearly 90% of cases, from IDH mutant GBMs, usually arising as secondary tumors from lower grade gliomas and otherwise not specified GBMs [2]. However, additional molecular markers maybe found in specific subsets of patients. The blood–brain barrier (BBB) further complicates GBM treatment, limiting the access to the tumor mass when anticancer molecules are systemically administered. Moreover, targeting GBM growth is complicated by the presence, within the tumor mass, of at least two populations of cancer cells, self-renewing tumor stem cells as well as more differentiated ones that respond differently to the various chemo- and radiotherapies, and whose biology is still not completely unraveled. Hence, both basic and clinical research need new tools for wrecking GBM conundrum.

Due to the peculiar biology of the parental viruses, retroviral-based vectors (gamma-retroviral and lentiviral vectors) display some interesting features, including persistent gene expression due to integration of their genome into the host DNA. In the last three decades, lentiviral vectors (LV) have been widely used in the field of GBM research, because of their advantages over gammaretroviral vectors, which were initially developed [3]. In particular, LV are more stable and less prone to insertional mutagenesis with respect to gammaretroviral vectors. Twenty-five years have passed since the earlier studies demonstrated that the Human Immunodeficiency Virus type 1 (HIV-1), the best-known lentivirus, could be modified for efficient gene delivery to lymphocytes [4,5] and to nondividing cells [6]. Currently, third-generation HIV-based vectors, which are highly improved in their transduction efficiency and safety, are adopted in several clinical trials [7]. LV offer several advantages over different viral and nonviral gene delivery vehicles: (i) the ability of transducing both dividing and resting cells; (ii) high efficiency in delivering transgenes to primary and stem cells; (iii) the capability of integrating their genome into the host DNA, leading to a sustained transgene expression; in addition, as mentioned above, LV show a potentially safer integration site profile when compared to gammaretroviral vectors [8]; (iv) a tissue tropism that can be modified by providing the viral core particles with an heterologous glycoprotein envelope. This process, known as pseudotyping, can be exploited to extend the natural tropism of the parental virus, which otherwise would be restricted to CD4 positive T cells and/or to allow an efficient and possibly specific targeting of cell types of interest. Relevant to GBM, LV have been successfully pseudotyped with the envelope of the lymphocytic choriomeningitis virus, resulting in a preferential transduction of tumor cells over the normal surrounding tissue [9]; (v) a low immunogenicity due to the virtual lack of viral protein expression after transduction; (vi) the ability to efficiently deliver not only transgenes, but also non coding (nc)-RNAs, such as small interfering RNAs (siRNAs) and microRNAs (miRNAs); and (vii) effective standardized and relatively easy to adopt methods for vector manipulation and production. These features, all together, led to the adoption of LV in different basic and translational studies. Indeed, LV are currently employed for (i) gene silencing by RNA interference (RNAi) with the aim of both analyzing the biological role of a certain transcript or for its knock-out in therapeutic approaches; (ii) transgene overexpression in several cell types, including primary and stem cells, as well cell marking and tracing, in vitro and in vivo; (iii) strategies aimed at vaccine development; (iv) the generation of transgenic animal models; (v) the induction of pluripotent stem cells starting from adult cells (iPSCs); and (vi) genome editing purposes.

In the field of GBM research, LV were originally designed for gene and cell therapy, yet their use falls well beyond this initial application. Indeed, LV have been valuable tools for exploring the tumor biology and for dissecting functions of various cellular pathways and proteins, including their possible role as therapeutic targets. Furthermore, LV were adopted to create and develop GBM animal models that closely mimic the clinical proteiform manifestations of this tumor (Figure 1). In this review, we aim at summarizing how LV contributed to gain further insights in our understanding of GBM biology and to design new therapeutic strategies, by mainly focusing on studies performed in the animal model.

## 2. LV as Tools for Studying Tumor Biology and the Identification of Therapeutic Targets

Given the above described features, LV have been extensively used to explore basic biology of GBM. Indeed, LV have been mainly adopted to achieve effective gene silencing with the aim of exploring the role of different proteins and/or cellular pathways in crucial aspects of GBM development and progression, as well as resistance to chemotherapy and radiation.

RNAi is a natural process through which expression of a targeted gene can be knocked down with high specificity and selectivity. The silencing mechanisms either rely on the degradation of the target mRNA, as in the case of siRNAs and short hairpin RNAs (shRNAs), or on the suppression of mRNA translation, as induced by miRNAs. RNAi can be also accomplished by artificially delivering these small nc-RNA species to the target cells, through the adoption of viral as well as nonviral vectors. LV represent one of the most adopted transfer tools in the field of RNAi, being extremely efficient in transducing small nc-RNAs into different cell types, including primary and stem cells.

In the context of GBM research, LV versatility combined to RNAi potency have been widely exploited. For instance, LV have been valuable tools for the generation of shRNA libraries that allowed a genome-wide, high-throughput genetic screening for the identification of genes involved in tumor initiation, maintenance, and in cell growth inhibition (e.g., see the work by Thaker and colleagues [10]).

By adopting LV designed to transduce specific shRNAs (shRNA-LV), Li and coworkers showed the importance of the dopamine receptor D2 (DRD2) signaling pathway in GBM. Validating this result, DRD2 transcript and protein expression were found increased in clinical glioblastoma specimens when compared to matched non-neoplastic tissues [11]. On the other hand, Wanka and coworkers demonstrated, through LV-mediated shRNA suppression of p53 and of its downstream effector SCO2, that this pathway is involved in tumor resistance to hypoxia [12].

With the same strategy it was shown that the SNARE proteins, which are crucial for the functioning of the cellular machinery involved in the cell interaction with extracellular environment, and more specifically syntaxin1, are crucial for GBM proliferation [13].

LV-mediated expression of shRNAs has been also adopted to investigate the molecular mechanisms contributing to the invasive nature of GBM within the central nervous system (CNS), a feature that highly contributes to the poor prognosis of this tumor. By adopting this strategy, the role of the A Disintegrin And Metalloproteinase (ADAM) family of proteins in invasiveness of GBM cells has been unraveled. Specifically, ADAM-9 was identified as a mediator of tenascin-C-stimulated migration of a specific class of cancer cells, the so-called tumor initiating cells [14]. Tumor initiating cells (TIC), or glioblastoma stem cells (GSC), are defined as cells with self-renewal ability, tumor-initiating capacity, and ability to give rise to a more differentiated progeny. A fraction of these cells seems to be responsible not only for the onset and recurrence of GBM, but also for its characteristic resistance to currently employed treatment [15]. Auvergne and coworkers by LV silencing were able to show a role for the Protease Activated Receptor 1 (PAR1) in self-renewal of human TICs. Indeed, the knock-out of PAR1 expression led to a robust decrease in the tumorigenicity activity of these cells [15]. A similar role was found for the transcriptional modulator High Mobility Group AT-Hook 2 (HMGA2), which is involved in motility and self-renewal of normal and cancer cells. When this factor was targeted by shRNA-LV in a GBM neurosphere cell line (HSR-GBM1), a reduction in cell stemness, invasion capacity, and ability to induce tumors was observed [16]. Furthermore, shRNA-LV allowed the identification of the protooncogene protein tyrosine phosphatase SHP2 as a positive factor for GSC proliferation and tumorigenicity [17].

Different proteins have been connected to GBM cell proliferation by LV-mediated silencing, among these, the fatty acid regulator hydroxysteroid dehydrogenase-like 2 (HSDL2) [18], as well as the ribosomal protein s15a [19].

The tumor microenvironment is greatly involved in driving tumor growth through proinflammatory and tumorigenic molecules that enable communication between tumor and nontumor cells, as a result of NF-κB- and/or AP-1-induced genes. Upstream of these molecules is TRAF3 Interacting Protein 2, which was silenced via LV-mediated shRNA, resulting in a reduced growth of GBM [20].

The cytoskeletal dynamics and angiogenesis are two factors playing a significant role in tumor formation/progression. In the first case, there is a great interest towards the identification of proteins involved in the microtubule formation. Indeed, these cell factors would represent suitable therapeutic targets, given that most of the already approved and in use antitumoral drugs interfere with this process. Stathmin, a 17 kDa regulator of microtubule dynamics, was recently targeted via LV-shRNA in GBM cells, resulting in a reduction of tumor formation and growth in nude mice [21]. Importantly, a recent study employing a new murine model of platelet-derived growth factor receptor-α (PDGFRα)-driven GBM, demonstrated that PDGFRα activity synergizes with the microtubule stabilizer drug vinblastine via stathmin dephosphorylation. This work reveals a connection between vinblastine cytotoxic effect on GBM cells and stathmin, prompting additional investigations on this protein as therapeutic target [22]. As far as angiogenesis is concerned, the role of the endothelium and its interaction with GBM cells have been explored by LV-shRNA, disclosing the role of programmed cell death protein 10 (PDCD10) dysregulation in tumor angiogenesis and in GBM progression [23]. Furthermore, downmodulation by LV-mediated silencing of the vascular endothelial growth factor (VEGF)/VEGF receptor (VEGFR) signaling pathway allowed its association to decreased tumor size coupled to an increased tumor necrosis in orthotopic glioma xenograft models [24]. In a recent study, the methionine aminopeptidase MetAP2 was LV knocked down in tumor cells resulting in reduced angiogenesis and tumor growth, positively affecting survival of mice [25].

Finally, LV transduction of shRNAs were used in the identification of intracellular targets positively correlated with longer patient survival, like phosphodiesterase 5 [26] and thioredoxin-interacting protein [27].

Currently, LV are also adopted as platforms to deliver the novel genome editing machinery, such as CRISPR/Cas9, for the knock-out of specific genes with potential roles in GBM formation and development. In this context, LV have been mainly adopted for the development of therapeutic strategies, an aspect that will be further discussed in the next paragraph of this review.

Even though gene silencing represents one of the main applications of LV in the study of GBM biology, these vectors have been also exploited for their ability to lead to a stable and sustained overexpression of factors potentially involved in peculiar features of this tumor. Interestingly, in dissecting the role of HMGA2 in promoting GBM tumorigenicity, Kaur and colleagues not only silenced its transcript by shRNA-LV, as described above, by they also overexpressed HMGA2 in the HSR-GBM1 neurosphere cell line, further validating RNAi results [16]. Data from patients show that mutated isocitrate dehydrogenase 1 is associated with better prognosis. By overexpressing this protein in glioma stem cells by mean of an ad hoc designed LV, it was proven that GBM cell proliferation, migration and invasiveness were reduced. Identified mechanisms giving reason of this result were the induction of apoptosis and cell differentiation accompanied by a down-regulation of the Wnt/beta-catenin signaling pathway [28]. Furthermore, LV-mediated overexpression led to the identification of the SERCA Ca2^+^-ATPase ATP2A2 as a beneficial factor for survival [29].

Finally, LV ability to efficiently transduce a wide range of cell types, including primary cells and cells in live organisms, has been exploited with the aim of dissecting the complex landscape of GBM protein expression patterns. In particular, Manricque and coworker adopted LV to optimize the methodology known as massively parallel reporter assay (MPRA) that facilitates the systematic analysis of transcriptional regulatory elements [30]. Thanks to the developed platform, those authors were able to demonstrate that local DNA sequence and regional chromatin affect regulatory activity, further advancing our vision of noncoding genome in GBM development [31].

## 3. LV as Tools for Gene and Cell Therapy of GBM

Due to its characteristics of aggressiveness, recurrence and resistance to traditional therapies (both chemo- and radiotherapy), GBM is considered among solid cancers one of the most suitable targets for innovative therapeutic approaches as gene and cell therapy. Indeed, GBM gene therapy history parallels gene therapy history itself, starting with approaches aimed at the delivery of either suicide or tumor-suppressor genes to the cancer mass, followed by more innovative strategies driven by the increased knowledge of the tumor biology. First of all, as already discussed in the previous paragraph, several proteins and cellular pathways were discovered to play crucial role in different aspects of GBM pathogenesis, and were targeted by gene silencing and, more recently, by genome editing approaches. In 2013, the journal Science selected as the breakthrough of the year cancer immunotherapy [32], a therapeutic strategy made possible by improved understanding of mechanisms accounting for the already well-known ability of cancer cells to evade the immune system control [33]. Viral vectors are employed also in immunotherapy approaches. Indeed, the idea of overcoming the immune suppression typical of the tumor microenvironment to increase the chance of therapeutic success was already exploited in earlier cancer gene therapy protocols, by combining immunomodulatory factors, as certain cytokines, to suicide proteins. In the case of GBM, for instance, Palù and coworkers generated a bicistronic retroviral vector expressing one of the most exploited suicide proteins, the thymidine kinase of herpes simplex virus type 1 (HSV1-tk), along with interleukin 2 (IL-2), which is widely adopted in cancer immunotherapy approaches. The developed vector was tested also in patients [34,35]. Currently, one of the most promising approaches of cancer immunotherapy is represented by viral vector mediated genetic modification of T cells to express chimeric antigen receptors (CAR) directed against specific cancer antigens [36].

Based on their features, LV have been widely used in gene and cell therapy approaches of GBM. In this context, LV were initially adopted mainly to deliver suicide genes, such as HSV1-tk. This viral protein works by activating through phosphorylation the prodrug ganciclovir, which, once activated, can block DNA replication leading to cell death. In the case of GBM, such a strategy resulted in significant tumor reduction (e.g., see [37,38]). Interestingly, the success of this approach was linked to a consistent bystander effect due to the presence of gap junctions among GBM cells [39].

Next, several preclinical and clinical studies adopted RNA interference as a strategy to treat GBM, especially by means of LV delivered shRNAs [40,41], targeting genes and pathways crucial for GBM biology, as previously described.

In addition to shRNAs, in the last years miRNAs have gained particular attention. Normally, these nc-RNAs are endogenously expressed and induce translational silencing/degradation by binding to target transcripts [42]. In this way, miRNAs play important regulatory roles both in physiological and in pathological conditions, cancer included. Indeed, almost all tumors are characterized by a peculiar pattern of miRNA expression that differs from the one found in the corresponding healthy tissue. This feature is the results of several complex mechanisms, such as an altered functioning in the cancer cell of the cellular machinery involved in the miRNA biogenesis [43]. Consequently, upregulation of oncogenes and/or downregulation of tumor suppressors occur that contribute to the development of tumors. GBM is not an exception, and different studies in the past years have clearly shown how miRNAs contribute to the phenotypic diversity of GBM subtypes and can be used as diagnostic and prognostic biomarkers, as well as therapeutic targets [44]. Specific miRNAs have been associated to the increase in GBM cell proliferation, resistance to apoptosis and cell death in general, invasiveness, induction of angiogenesis, and resistance to traditional treatments [45]. Not surprisingly, LV were used to investigate the involvement of specific miRNAs in different aspects of GBM biology. For instance, both overexpression and downregulation were obtained with an ad hoc designed LV, leading to the identification of miR-100 as a protective factor via interactions with the Fibroblast Growth Factor Receptor 3 (FGFR3) signaling pathway [46]. On the other hand, miR-297 was unraveled as a factor promoting survival of GBM cells, by targeting diacylglycerol kinase alpha [47].

Considering their crucial roles in tumor biology and their peculiar transcriptional signature in cancer cells, miRNAs have been considered a suitable target for developing novel therapeutic strategies to fight GBM. Antisense oligonucleotides and the so-called “miRNA sponges” [48] were used in strategies aimed at downregulating miRNAs playing a role in carcinogenesis [49]. In this context, LV have been adopted as a delivery system to achieve an affective and sustained expression of the anti-miRNA molecules also in vivo. For instance, Chen and coworkers adopted a LV to express a miRNA sponge able to block miR-23b function as an oncogene in GBM. Those authors showed that both in a glioma cell line and in an orthotopic mouse model miR-23b inhibition resulted in a significant reduction in tumor malignancy and angiogenesis, as well as in the invasiveness capacity of the tumor cells. As a consequence, cancer progression was affected [50]. On the other hand, the differential expression of specific miRNA in GBM cells with respect to healthy cells was exploited to achieve a targeted expression of suicide genes delivered by LV only within the cancer. Skalsky and coworkers generated an LV expressing HSV1-tk under the transcriptional control of miR-128, which is downregulated in GBM, obtaining a selective killing of cancer cells upon transduction and ganciclovir administration [51].

More recently, gene therapy approaches aimed at interfering with GBM biology have shifted on the use of the novel genome editing technologies. In particular, the Clustered Regularly Interspaced Short Palindromic Repeats (CRISPR) and CRISPR-associated (Cas) 9 system is currently one of the most exploited tools [52] and, as mentioned above, LV are among the most suitable viral vectors for its efficient delivery to several target cells in vitro and in vivo. Under this respect, Tome-Garcia and coworkers have achieved a strong impairment of cancer cell migration by LV-mediated transduction of an ad hoc-designed CRISPR-Cas9 system [53]. On the other hand, in a very recent study, Sun and coworkers demonstrated that GBM growth can be impaired increasing mice survival, by editing the sequence encoding the vascular laminin-411 that is overexpressed in higher grade GBM [54].

Finally, gene therapy approaches for the treatment of GBM based on LV vectors have also exploited their ability to lead to a sustained expression of heterologous genes in vitro and in vivo and their versatility to allocate expression cassettes. As an example, Sanchez-Hernandez and colleagues have recently designed and developed a LV to obtain equimolar expression of the Growth Arrest Specific 1 (GAS1) and of the tumor suppressor phosphatase and tensin homolog (PTEN). Indeed, the aim of the study was to investigate the potential additive effect of these proteins on cancer cell proliferation, considering that GAS1 is known to induce apoptosis in GBM cells, while PTEN blocks the phosphatidylinositol 3-Kinase (PI3K)/protein kinase B or the Akt pathway. To achieve a similar expression level of both proteins, the authors exploited the p2A peptide–base expression system, which has been demonstrated to work well in the context of LV, to allow the expression of both proteins under the transcriptional control of the same promoter. In this way, proliferation of GBM cells was significantly affected in vitro and upon LV inoculation into immunosuppressed mice [55].

An interesting application of LV as tool to overexpress proteins that might help in treating GBM was set up by Lamb and coworkers. The GBM standard therapy in clinical application is based on a combination of temozolomide with radiation therapy. Temozolomide (TMZ) is an alkylating agent that binds to DNA and interferes with replication, leading to breaks within the DNA and, as a consequence, cell death. However, TMZ has a limited activity mainly due to the overexpression in cancer cells of the DNA repair protein O6-methylguanine-DNA methyltransferase (MGMT). On the other hand, it is known that γδ T cells, a subset of T cells, that are able to recognize stress-associated Natural Killer Group 2D (NKG2D) receptor ligands expressed by GBM and efficiently reduce tumor expansion. Noteworthy, TMZ induces the expression of these ligands on TMZ-resistant GBM cells making them more vulnerable to γδ T cells. However, TMZ is also toxic to γδ T cells. To overcome this issue, γδ T cells were transduced with LV expressing MGMT, thus conferring them resistance to TMZ. Transduced γδ T cells maintained their biological characteristics and displayed cytotoxicity toward TMZ resistant GBM cell lines, upon administration of TMZ, thus fostering their use to treat GBM [56].

### 3.1. LV-Based Gene Therapy Approaches Aimed at Overcoming GMB Resistance to Therapy

One of the peculiar features of GBM is the onset of resistance to both radiation and chemotherapy which leads to rapid recurrence after treatment.

By generating LV expressing an anti-MGMT shRNA, Viel and coworkers observed a specific inhibition of the MGMT expression in GBM cell lines as well as in the animal model. Furthermore, combining the developed LV with TMZ treatment those Authors were able to show a significant reduction in tumor formation [57].

In addition to MGMT, p53 is known to be involved in GBM resistance to TMZ [58]. Furthermore, protein phosphatase 1D magnesium-dependent PPM1D, a member of the protein phosphatase 2C (PP2C) family, seems to play a role as well by modulating p53 function [59]. Wang and colleagues showed that silencing of PPMD1 by LV-mediated expression of a specific shRNA improved the effect of TMZ by inducing apoptosis of GBM cells mainly through the PIK3R1/AKT pathway [59]. More recently, starting from the observation that SRY-Box 9 (SOX9) protein expression is linked to poor prognosis in GBM patients, its encoding transcript was targeted via shRNA-LV, demonstrating that this protein along with carbonic anhydrase 9 (CA9) are part of an oncogenic pathway. Furthermore, this study clearly showed that the inhibition of such a pathway leads to an increased sensitivity of GBM to TMZ [60].

On the other hand, not only resistance to TMZ is a major problem in GBM therapy, but also radioresistance contributes to scarce efficacy of therapy. The interaction of various proteins with radiation therapy has been exploited using LV. Shi and coworkers investigated the effect of K5, a kringle domain of plasminogen known to induce apoptosis of dermal microvessel endothelial cells, combined with the positive feedback effect of the Early growth response-1 (Egr1) promoter on sodium/iodide symporter (NIS) gene upon ^131^I radiation. To this end, they generated a LV expressing K5 under the transcriptional control of the cytomegalovirus promoter and NIS under the control of the Egr1 promoter. The GBM cell line U87 was transduced with the developed LV and employed to induce tumor formation in nude mice. In this model, K5 and irradiation showed an addictive effect on glioma growth, cell proliferation and capillary density [61]. Oxidative stress may interact with cell survival: when Nox4 NADPH oxidase, which contributes to generate reactive oxygen species in GBM cells, was knocked-down via LV, tumor proliferation, invasion and radioresistance were diminished [62]. Stably expressing low-represented miRNA MiR-224 by means of LV was also shown to increase sensitivity of GBM cells to radiations [63]. Finally, the transcript encoding ATM protein, which is mutated in ataxia-telangiectasia development conferring radiosensitivity to the patients, was silenced by LV-mediated RNAi increasing tumor response to radiotherapy [64].

Interestingly, GSCs seem to play a crucial role in GBM resistance to both radio and chemotherapy due to cellular and microenviromental mechanisms [65,66,67]. Interestingly, the possibility to pseudotype LV offers the opportunity of specifically targeting a subset of GSCs that express on their surface the glycosylated pentaspan transmembrane protein CD133. Specifically, Bayin and coworkers were able to generate a LV displaying a single chain antibody against CD133 on its envelope. Those authors showed that such a LV selectively transduced CD133-expressing cells in GBM xenografts in NOD SCID mice, while sparing normal brain tissue [68]. This LV represents a valuable tool to dissect the role of CD133 positive GCSs in tumor biology in general and in its resistance to chemotherapy in particular. Furthermore, it can have an application in gene therapy approaches aimed at targeting this subset of cells in particular.

### 3.2. Immunotherapy of GBM via LV Expressing CAR Specific to GBM Antigens

As mentioned above, immunotherapy of cancer is emerging as one of the most promising strategies to fight cancer in general and GBM in particular. Among immunotherapy approaches, one of the most efficient is represented by the injection of genetically modification of autologous T cells to express chimeric antigen receptors (CARs) with defined specificities for tumor-associated antigens independent of MHC restriction. Typically, CARs are composed of domains of synthetic antibodies recognizing the selected antigen along with intracellular activation/stimulatory domains that transduce signals from the surface receptor inside the cell. Different generations of CARs do exist that differ in the number of intracellular stimulatory domains. First-generation CARs contain a single activation domain (i.e., the T cell receptor ζ chain CD3ζ). Second- and third-generation CARs consist of one or two additional costimulatory signaling domains, such as CD27, CD28, CD134 and CD137, leading to an increase in activated T cell overall survival, proliferation and persistence [69]. This innovative cell therapy approach has been shown to be particularly active in the case of hematological malignancies, while the results for solid tumors have not been as encouraging, likely due to their specific immunobiological features [70]. Interestingly, clinical studies have shown the safety and feasibility of this type of cell therapy in the treatment of GBM and signs of success at least in a subset of patients [71]. Different CARs directed against antigens expressed on the GBM cell surface have been generated and tested. In particular, major exploited targets include so far, the epidermal growth factor receptor variant III (EGFRvIII), the human epidermal growth factor receptor 2 (HER2), the IL13 receptor alpha 2 (IL13Rα2) and ephrin type A receptor 2 (EphA2). CAR T cell therapy against GBM targeting EGFRvIII and HER2 are already in clinical trials [72].

Even though retroviral vectors have been traditionally used in approaches of CAR therapy of GBM, LV were also used in this context. For instance, Kuramitsu and colleagues constructed a LV expressing a CAR directed against the EGFRvIII, to transduce CD3 positive (+) T cells, rendering them able to specifically lyse EGFRvIII positive GBM cells. Importantly CAR-expressing CD3(+) T cells reached intracranial xenografts of GBM in mice [73]. These results were further expanded by targeting both the mutated and the wild type (wt) form of EGFR. Specifically, natural killer (NK) lymphocytes were transduced with a LV expressing a second generation CAR targeting both wild type EGFR and EGFRvIII. In detail, the human NK cell line NK-92 was transduced and adopted to intracranially inject two orthotopic xenograft mice models, achieving efficient suppression of tumor growth and prolonged animal survival [74]. More recently Murakami and coworkers generated a novel type of NK cell line (KHYG-1) transduced with a LV to express a CAR targeting ECGRvIII. The authors were able to establish the transformed cell line and to show that it inhibited GBM cell growth via apoptosis depending on EGFRvIII expression [75].

On the other hand, LV have been also used to express CAR directed towards novel GBM specific antigens. In this context, Shiina and coworkers generated a LV expressing a third-generation CAR specifically recognizing podoplanin, a transmembrane mucin-like glycoprotein, expressed in the mesenchymal GBM subtype. The developed LV was used to obtain CAR-T cells that were active against podoplanin-positive GBM cells in vitro. Furthermore, upon injection of transduced T lymphocytes in immunodeficient mice, a significant growth inhibition of GBM xenograft was observed [76].

### 3.3. LV Transduced Cells as Carriers of Therapeutic Molecules for the Treatment of GBM

A different approach to cell therapy takes advantage of specific cell types that can be used as carriers for delivering therapeutic genes directly to the cancer mass. Different studies have clearly demonstrated that stem cells, in particular neural stem cells (NSCs) and mesenchymal stem cells (MSCs), have a tropism for brain tumors [77,78]. This feature has prompted the adoption of stem cells as vehicles for targeted therapeutic molecule delivery [79]. One of the main obstacles to the delivery of both viral vectors and stem cells to a tumor mass located within the CNS is represented by the BBB. Even though the direct injection into the brain works, the associated risks and the procedure complexity might render this method difficult to apply in clinical trials and eventually in a clinical setting. Much work has been carried on finding the best route of modified stem cells administration to overcome the blood–brain barrier [80,81]. In the case of GBM, intranasal delivery of cell carriers appears to be an efficient method [82]. Furthermore, LV, for their capacity of transducing stem cells, have been widely adopted as viral vectors to transform both MSCs and NSCs, also in the context of GBM cell therapy. In 2011, Balyasnikova and colleagues published a paper focused on the adoption LV-engineered NSCs to express membrane-bound tumor necrosis factor-α-related apoptosis-inducing ligand (NSCs-mTRAIL). Indeed, while it is well known that TRAIL can selectively induce apoptosis in GBM cells, some tumors were found resistant to the soluble form of this proapoptotic agent. The authors showed that NSCs-mTRAIL, improved the survival of mice bearing intracranial glial tumor xenografts [83]. A very interesting study, fully exploiting LV versatility, was carried on by Bagò and coworkers [84]. In their work these authors adopted LV both to set up a novel strategy to rapidly converting human skin fibroblasts into tumor-homing early-stage induced NSCs, named h-iNSC^TE^. Next, they further adopted LV to engineer h-iNSC^TE^ to express reporter proteins and different therapeutic molecules. Finally, they showed the tumor-homing migration and therapeutic efficacy of transformed h-iNSC^TE^ in two different mouse models of GBM, one based on orthotopic established GBMs and the other on implantation of patient-derived cancer lines [84]. Specifically, stem cells transformed with LV expressing either TRAIL or HSV1-tk significantly affected the size of GBM xenografts and displayed a positive effect on animal survival. Furthermore, the authors were able to show that delivery of HSV1-tk expressing h-iNSC^TE^ into the postoperative surgical resection cavity delayed the regrowth of residual GMB in mice. Altogether these data demonstrate that h-iNSC^TE^ might represent a valid platform to generate tumor-homing cytotoxic cells to be applied for the treatment of GBM.

LV have been used to efficiently transduce also MSCs leading to a sustained expression of the transgene of interest. In the context of GBM, Bak and coworkers employed MSCs derived from human embryonic stem cells to deliver HSV1-tk to the tumor, showing that the transduced cells were able to migrate from the site of inoculum in one cerebral hemisphere towards the cancer located in the opposite side of the mouse brain, delivering the suicide gene [85].

LV-expressing cytosine deaminase (CD) or enhanced green fluorescent protein (eGFP) was constructed and transduced into rat MSCs (MSC-CD/eGFP). In addition to HSV1-tk, CD is another widely exploited suicide protein that activates the prodrug 5-fluocytosine into a cytotoxic agent, the 5-fluorouracyl (5-FU), which is phosphorylated inside the cells. The active metabolites can be incorporated into DNA and RNA resulting in the inhibition of their synthesis and cell death. As for HSV1-tk based strategy, also in this case, gap junctions mediate the bystander killing of tumor cells. The LV-mediated expression of the reporter protein allowed the tracing of the transduced MSCs in vivo, upon their intracranial injection in a GBM rat model. The cells were found to localize mainly at the interface between cancer and normal tissue. After administration of 5-FU, MSC-CD/eGFP determined a significant decrease in the tumor mass with increased cancer cell apoptosis and mice survival [86]. Balyasnikova and colleagues adopted LV engineered human bone marrow derived MSCs to express m-TRAIL. The authors were able to show that MSCs can migrate into the brain after nasal instillation and infiltrate intracranial glioma xenografts in a mouse model. Furthermore, MSCs expressing m-TRAIL increased the survival of mice treated with a combination of m-TRAIL expressing MSCs and irradiation [82]. An abundant and accessible source for MSCs is the adipose tissue. De Melo and colleagues succeeded in purifying MSCs from this source and transduced them with an LV expressing HSV1-tk. The Authors demonstrated that GBM cells died when co-cultured with these cells and upon GCV treatment. More importantly, they showed that when injected into the brains of nude mice bearing GBM, LV-transduced MSCs were chemoattracted to the tumor and killed cancer cells in situ, via a bystander effect, without significantly affecting normal brain tissue [87].

In an interesting approach, instead of NSCs or MSCs, patient-derived olfactory ensheathing cells (OECs) were modified by LV transduction to carry HSV1-tk to primary cultured human GBM, taking advantage of OECs natural tropism to the brain. Furthermore, these cells should reduce technical and ethical problems, being obtained from the olfactory mucosa, a source easily accessible from the patients themselves. The study showed that transduced cells affected survival of tumor cells after ganciclovir administration. This finding opened up the possibility to exploit OECs as vehicle to transfer anticancer molecules to brain tumors [88]. Indeed, more recently, Carvalho and coworkers studied OEC application in a cytotoxic therapeutic approach of GBM in vivo. Specifically, they transduced OECs with an LV-expressing CD fused in frame to the uracil phosphoribosyltransferase (CU) that should enhance the overall antitumor effect [89]. By injecting patient-derived GSCs in the striatum of nude mice brains, the authors were able to show that OECs migrated from the nasal cavity to the tumors, reached infiltrative GBM cells, and delivered the fusion protein, affecting tumor growth and mice survival. This work represents the first demonstration that OECs can efficiently target GBM in vivo and deliver therapeutic transgenes upon intranasal delivery [90].

Finally, a very interesting recent study demonstrated that it is possible to use genetically engineered macrophages to drive the expression of secreted proteins that can influence the GBM microenvironment, enhancing the immune system recognition of cancer cells. Importantly, to this end, a novel LV expression system was designed and tested that allowed the modification of macrophages [91]. This study clearly demonstrates the versatility of LV and their feasibility in approaches that require vectors easy to manipulate and efficient in the transduction of primary cells.

## 4. LV to Create Animal Models

One of the major challenges in studying GBM is the difficulty of reproducing in the laboratory all the features of this tumor, that has no clear molecular signature and whose effects may vary also according to the site of origin within the brain.

Different experimental GBM models are available in both invertebrates and vertebrates, from chemically induced to xeno- and allograft as well as viral mediated or genetically engineered animals [92], for the study of tumor biology and the development of new treatments. Despite easy manipulation and low-cost, few models have been created in Drosophila by manipulating glial progenitor cells [93]. More options are available for the zebrafish *Danio rerio*, which allows an easy tracing of single cells within the tumor mass induced via xenograft in animals [94], or in tumors originated by CRISPR/Cas9 genetic tailoring [95]. A model closer to human is the dog, which can develop spontaneous brain tumors with histological features similar to human counterpart, but ethical reasons and long timeframe for tumor development hamper its wide use as a model for GBM studies [96]. The most widely used animal models for the study of GBM are generated in rodents, which allow flexible and fast manipulations in a complex brain system. Initially, GBM-like tumors were induced in rodents by ethylnitrosurea injection or methylnitrosurea administration, but the complexity and low reproducibility of the procedure lead to its discontinuation [97]. Next, allograft and xenograft models were established, with the limits that the initial stages of tumor development are different from naturally occurring disease. Moreover, in the case of xenograft, immunodeficient mice are required so that tumors escape the immune destruction step, a phase taking place in naturally occurring cancers. Recently, however, an interesting approach was designed for mice orthotopic tumors, in which primary cells from patients or GBM cell lines were transduced with LV expressing green fluorescent protein (GFP)/firefly luciferase fusion proteins before implantation in the frontal lobe of mice. These proteins were detected in vivo via bioluminescence imaging [98], to allow the monitoring of GBM development also during chemotherapy administration.

LV were used in immunocompetent mice to overexpress oncogenes under Cre-loxP control, inducing tumors in some brain regions of the adult mouse: the expression in specific cell types or regions (subventricular zone or hippocampus) of the activated forms of different oncoproteins such as H-Ras and AKT successfully induced GBM growth [99,100], pointing at LV overexpressing these factors under the control of Cre-loxP as a powerful tool for GBM generation. Similarly, LV have been adopted to express Cre recombinase in mice carrying conditional alleles for different oncogenes and oncosuppressors successfully generating high-grade gliomas (e.g., see the work by de Vries and colleagues [101]). In a very elegant work, Niola and colleagues combined LV-mediated overexpression of oncogenic H-Ras, silencing of the oncosuppressor p53 and Cre-loxP technology to generate a mouse model of high-grade GBM and to study the role of Inhibitor of Differentiation (ID) proteins in TIC biology and tumor maintenance [102].

More recently, recombinant LV expressing a shRNA targeted to the Cyclin-dependent kinase Inhibitor 2A (Cdkn2a) and expressing the Platelet-Derived Growth Factor subunit B (PDGFB) were generated to induce proneural-type GBM growth in the dentate gyrus of immunocompetent mice, setting up an easy procedure for preclinical testing which does not require genetically engineered mice [103]. This model is powerful since it uses different version of the vector, for the deletion of floxed genes in transduced cells or for eGFP visualization, by direct injection into the dentate gyrus of immunocompetent adult mice, inducing specific gene expression signature reminiscent of proneural GBM. The rapid development of the tumor allows preclinical testing of drugs in immunocompetent animals.

Lynes and coworkers performed intracranial injection of LV expressing different factors involved in the receptor tyrosine kinase (RTK)/RAS/PI3K pathway to induce high-grade gliomas in rats, not only obtaining useful animal models, but also elucidating the role of the selected proteins in GBM development [104].

Lastly, a GBM model was obtained in the tree shrew *Tupaia belangeri*, which is phylogenetically closer to primates than rodents: a LV was used to transduce mutant H-Ras and a shRNA to silence p53, resulting in the fast development of tumors with histological features reminiscent of GBM, including aggressiveness and GBM histological features [105]. The genetic analysis suggested similarity to the mesenchymal subgroup of human GBM, with a better match with respect to comparable mouse models.

## 5. Conclusions

Currently, LV represent a valid tool to explore GBM biology, via the creation of improved animal models and the selective targeting of cellular specific pathways, and also in the complex tumor microenvironment. They can also support the design of innovative therapies, at the preclinical and clinical level (Table 1). Challenges and drawbacks still have to be faced in the development of safe and effective molecular approaches for GBM treatment. One of the main issues is represented by the difficulty for the treatments to gain access to the tumor mass due, for example, to the presence of the BBB. Furthermore, the need of highly specific therapeutic approaches combined with the biological characteristics of the CNS cell repertoire, complicate GBM therapy. Thus, the availability of versatile gene delivery systems, allowing, for instance, the targeting of selected cell types (pseudotyping) and highly efficient in transducing primary and stem cells, is extremely useful. In this context, LV, due to their peculiar features, are among the best vectors to be adopted in gene and cell therapy protocols based on the expression of small noncoding RNAs and on the engineering of primary cells to express CARs or to become carriers of therapeutic molecules to the tumor mass.

## Figures and Tables

**Figure 1 cancers-11-00417-f001:**
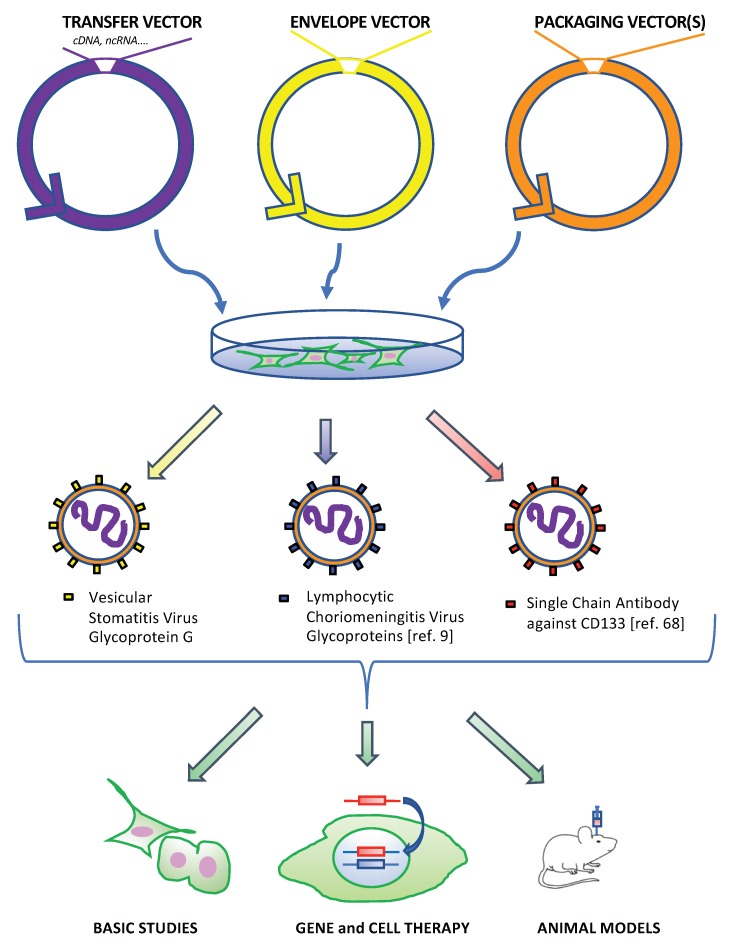
Overview of the main applications of lentiviruses LV in glioblastoma (GBM) research. Once transfected in appropriate cell lines, the packaging and envelope plasmids express the viral structural and enzymatic proteins (packaging vector(s)) along with the envelope glycoprotein (envelope vector) leading to the formation of viral particles that will incorporate the transgene encoding vector (transfer vector). The vesicular stomatitis virus glycoprotein G (VSV-G) is a widely employed envelope which confers to the recombinant particles the ability to infect a large range of target cells, including primary and stem cells. Additional envelopes can be adopted with the aim of restricting vector entry to cell types of interest, as explained in the text. In the context of GBM research, recombinant lentiviral particles were used in basic studies focused on tumor biology/new therapeutic target discovery, in gene and cell therapy approaches and to generate suitable animal models.

**Table 1 cancers-11-00417-t001:** Summary of the studies discussed in the text.

LV Main Applications in GBM Research	References
**Basic Studies**	
Dissecting the tumor biology	[10,11,12,13,14,18,19,23,26,27,28,29,30,31]
Identification of novel therapeutic targets	[15,16,17,20,21,22,24,25]
**LV as Tools for Gene and Cell Therapy of GBM**	
Gene Therapy	
Gene therapy approaches based on transgene expression	[37,38,39,55,56]
Gene therapy approaches based on silencing/gene editing	[46,47,50,51,53,54]
Gene Therapy approaches aimed at overcoming GMB resistance to therapy	[57,58,59,60,61,62,63,64,68]
Immunotherapy of GBM via LV expressing CARs	[72,73,74,75,76]
Cell Therapy	
LV transduced cells as carriers of therapeutic molecules	[82,83,84,85,86,87,88,90,91]
**LV to Create Animal Models**	
Murine models	[97,98,99,100,101,102,103,104]
Non-murine models	[93,94,95,96,105]

LV: Last-generation lentiviruses; GBM: Glioblastoma; CAR: chimeric antigen receptors.

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
