# Peer review of "Lentiviral Vectors as Tools for the Study and Treatment of Glioblastoma"

_cancers, 2019, doi:10.3390/cancers11030417_

Reviewer 1 Report

In the review entitled ‘Lentiviral vectors (LV) as tools for the study and treatment of glioblastoma (GBM)’, the authors describe the contribution of LV in the understanding of GBM biology and LV-therapeutic options.

This review is well-written and well-documented.

Main concerns

- Unfortunately, I found this review difficult to follow. Adding one table/paragraph will certainly help the readers if each table contains the examples cited in the related paragraph.

- Figures to illustrate main examples and a general figure about the strategies described in the review, will be greatly appreciated

- The problem of the BBB passage is barely mentioned.

- The conclusion would benefit from a development of the sentence l491-492 ‘As several challenges and drawbacks still have to be faced in the development of…’

Minor point

- Paragraphs are incorrectly numbered.

Author Response

REVIEWER 1:

Comments and Suggestions for Authors

In the review entitled ‘Lentiviral vectors (LV) as tools for the study and treatment of glioblastoma (GBM)’, the authors describe the contribution of LV in the understanding of GBM biology and LV-therapeutic options.This review is well-written and well-documented.

 AUTHORS: We thank the Reviewer.

Main concerns

- Unfortunately, I found this review difficult to follow. Adding one table/paragraph will certainly help the readers if each table contains the examples cited in the related paragraph.

AU: we have added one new table which summarizes the references cited in the text (Table 1).

- Figures to illustrate main examples and a general figure about the strategies described in the review, will be greatly appreciated

AU: We have added a general figure to summarize the strategies described in the text.

- The problem of the BBB passage is barely mentioned. 

AU: we have now expanded the discussion on the blood-brain barrier at lines 33-35, 400-405, 540-543.

- The conclusion would benefit from a development of the sentence l491-492 ‘As several challenges and drawbacks still have to be faced in the development of…’ 

 AU: The conclusion has been expanded as requested (lines 540-546).

Minor point

- Paragraphs are incorrectly numbered.

AU: we checked the manuscript we submitted and found the paragraph were correctly numbered – as they are now in our computers. Some formatting has been done by the Editorial Office, probably there is a problem of compatibility with our computer.

Reviewer 2 Report

The review by Del Vecchio et al. gives a concise overview of applications for lentiviral vectors in basic science and treatment of glioblastoma. The manuscript would benefit from a figure and/or table to make it more appealing to the reader. The authors could for example provide an overview figure both for treatment and basic science, where lentiviral vectors are used.

Minors

"LV to create animal models" should be numbered with "4."

page 4, l. 196: "suicide factors, like HSV1-tk", should be replaced by "suicide genes, such as HSV1-tk".

page 6, ll. 269-270: "LV found application....." This sentence makes no sense.

Author Response

The review by Del Vecchio et al. gives a concise overview of applications for lentiviral vectors in basic science and treatment of glioblastoma. The manuscript would benefit from a figure and/or table to make itmore appealing to the reader. The authors could for example provide an overview figure both for treatment and basic science, where lentiviral vectors are used.

AU: we have added a summary figure (Figure 1) and a table (Table 1).

Minors

"LV to create animal models" should be numbered with "4."

AU: See above, probably there was a compatibility problem. In our computer paragraphs appear numbered correctly.

page 4, l. 196: "suicide factors, like HSV1-tk", should be replaced by "suicide genes, such as HSV1-tk".

AU: The text has been modified as requested by the Reviewer.

page 6, ll. 269-270: "LV found application....." This sentence makes no sense.

AU: the text has been modified as requested. 

 Round  2

Reviewer 2 Report

Comments have been adequately addressed.